# Interactive Media for Understanding ML Methods: A Case-Study on Graph Neural Networks

**Ameya Daigavane, Balaraman Ravindran & Gaurav Aggarwal**
Google Research
Bangalore, India
{ameyasd,balaramanr,gauravaggarwal}@google.com

## Abstract

We demonstrate the advantages of an interactive medium for explaining the key principles and mathematical machinery behind graph neural networks. We discuss the challenges we faced while creating an expository article on this topic using interactive elements. Our exhibit is available at `https://distill.pub/2021/understanding-gnns/`.

## 1 Introduction

Graph neural networks (GNNs) have become extremely popular across a wide range of domains: relational modelling, physics simulations, knowledge-graph creation, model-based reinforcement learning, to name a few. Many GNN models, however, are inherited from the same fundamental principles and building blocks. To readers unfamiliar with the history of the field, this can make GNNs seem unintuitive.

Furthermore, visual descriptions of GNNs in existing publications, such as the examples shown in Figure 1, are typically non-descriptive and/or fixed to a particular simple graphs, due to the inherent limitations of the medium. This makes it unclear what exactly GNNs are computing, and how they relate to ML algorithms that readers are already likely to be familiar with, such as Convolutional Neural Networks.[1]

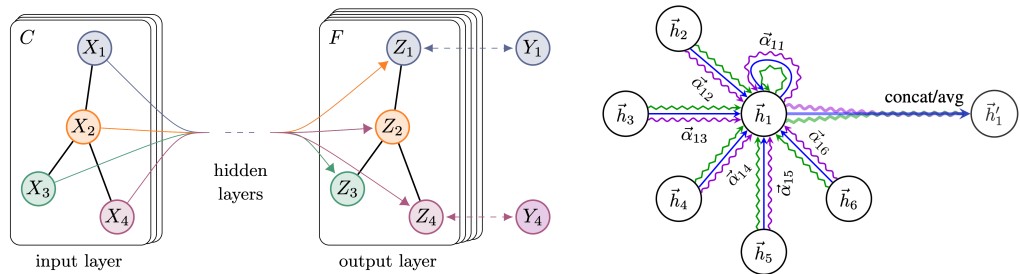

Figure 1: Examples of visual descriptions of the GCN model (left) from Kipf & Welling (2017) and the GAT model (right) from Veličković et al. (2018) as seen in 'traditional' research papers.

Significant effort has been developed to visualize classical graph algorithms such as breadth-first search, shortest paths and network flows[2], and even to visualize graphs themselves[3] with easily accessible implementations via tools such as GraphViz and NetworkX. However, to the best of our knowledge, interactive tools to describe what GNNs compute have yet to be developed.

---

[1]Many exemplary educative resources (such as Olah et al. (2017) and Carter et al. (2019)) already exist for CNNs and related methods.

[2]See, for example, VisuAlgo.

[3]See Gibson et al. (2013) for a comprehensive comparison of popular graph-layout algorithms.

To bridge this gap in understanding GNNs, we have created an expository article 'Understanding Convolutions on Graphs', which exploits the interactivity of the web format to visually explain the non-trivial concepts that drive GNNs. In this paper, we use this article as a running example to highlight the usefulness of interactive media to describe otherwise seemingly difficult concepts and the challenges involved. The rest of the paper is organised as follows:

- In Section 2, we discuss the specific visualizations we have designed, comparing them to 'static' visualizations seen in traditional research articles on GNNs.
- In Section 3, we emphasize the variety of challenges we faced while creating our exhibit. We expect these challenges to be encountered by authors of similar articles in the future, even beyond graph neural networks.
- In Section 4, we address concerns about the accessibility of our article's presentation.
- In Section 5, we conclude with a summary of the benefits and drawbacks of interactive media for explaining methods in machine learning, from our perspective.

Finally, to enhance the reproducibility of our interactive visualizations, we have created ObservableHQ notebooks in JavaScript[4], which can be forked and modified by anyone interested to create similar exhibits.

## 2 DESIGNING VISUAL DESCRIPTIONS OF GNNs

### 2.1 UNDERSTANDING SPECTRAL CONVOLUTIONS

Our article first discusses spectral convolutions, an operation that 'updates' the features of nodes within a given graph. Spectral convolutions are defined using eigenvectors of the graph Laplacian, $L$:

$$L = D - A.$$

We provide a visualization of these eigenvectors for a specific graph: the regular $16 \times 16$ grid, reminiscent of a 2D image with only $1$ channel, where each node is connected to either $3, 5$ or $8$ neighbours depending on its position within the grid. We chose a grid because:

- Readers are likely to be familiar with the structure of the grid.
- The grid is particularly easy to visualize just as an image, because nodes can be identified by their absolute position.

The resulting visualization (Figure 2) enforces the idea of how eigenvectors of the Laplacian are tied to 'smoothness' of node features across a graph, and how the spectral representation is the analogue of the Fourier representation for images in terms of underlying frequencies.

### 2.2 UNDERSTANDING CHEBNET

ChebNet (Defferrard et al., 2016) works with polynomial filters, defined as polynomials $p(L)$ of the graph Laplacian $L$. At first, however, it is unclear what these polynomials define, especially as their degree increases. We have created an interactive visualization (Figure 3) where the reader can change the coefficients of these polynomials, and visualize their effect on a regular grid. The reader can toggle node features, as well. The spectral representations of these filters are also depicted, highlighting the relation to spectral convolutions defined earlier.

### 2.3 INTERACTIVE GNN MODELS

To ensure our exhibit is indeed relevant to modern-day practitioners, we have first picked the most popular GNN models:

- Graph Convolutional Networks (GCN) (Kipf & Welling, 2017)

---

[4]Link to Observable notebooks can be found at `https://distill.pub/2021/understanding-gnns/#supplementary`.

- Graph Attention Networks (GAT) (Veličković et al., 2018)
- Graph Sample and Aggregate (GraphSAGE) (Hamilton et al., 2017)
- Graph Isomorphism Network (GIN) (Xu et al., 2019)

and then created annotated formulas for each of them, using a color scheme as shown in Figure 4.

To further strengthen the reader's understanding, we have created interactive versions of these models, where readers can see update equations at a node while varying the model's parameters. Readers can select their node of interest by clicking on it: the formulas and resulting calculations adapt automatically in real-time (Figure 5). They can additionally randomize the graph structure and features to see how these equations change.

Compared to the static diagrams shown in Figure 1, our visualizations are not only flexible with respect to the graph topology, but also detailed with the exact computation performed by the GNN model at the different nodes, without being overly complicated.

## 2.4 EXPERIMENTS WITH THE GAME OF LIFE

Our article experiments with GNNs for Conway's Game of Life (Gardner, 1970), comparing them to a CNN baseline. While these models were trained on a fixed set of example grids, we expose these models to user-inputted grids by running them in-browser. This visualization (Figure 6) allows readers to explore different types of errors that these models make; by simply clicking on the input grid, the reader can see predictions change in real-time.

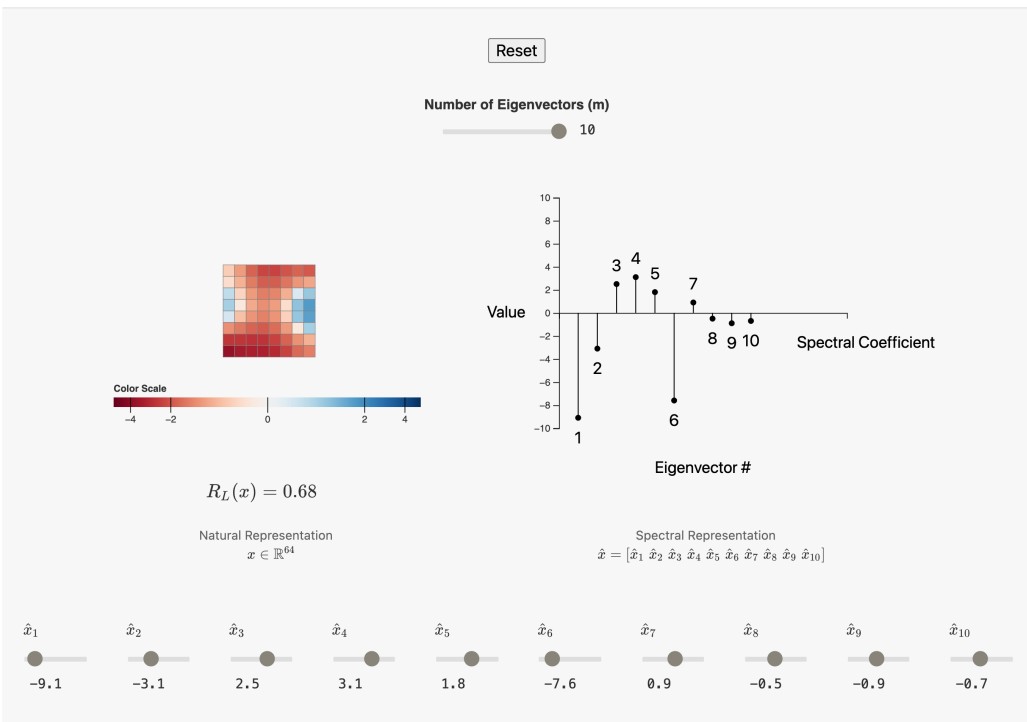

Figure 2: Readers can change the sliders representing spectral coefficients (shown on the right) to see how the actual features on the grid (shown on the left) change.

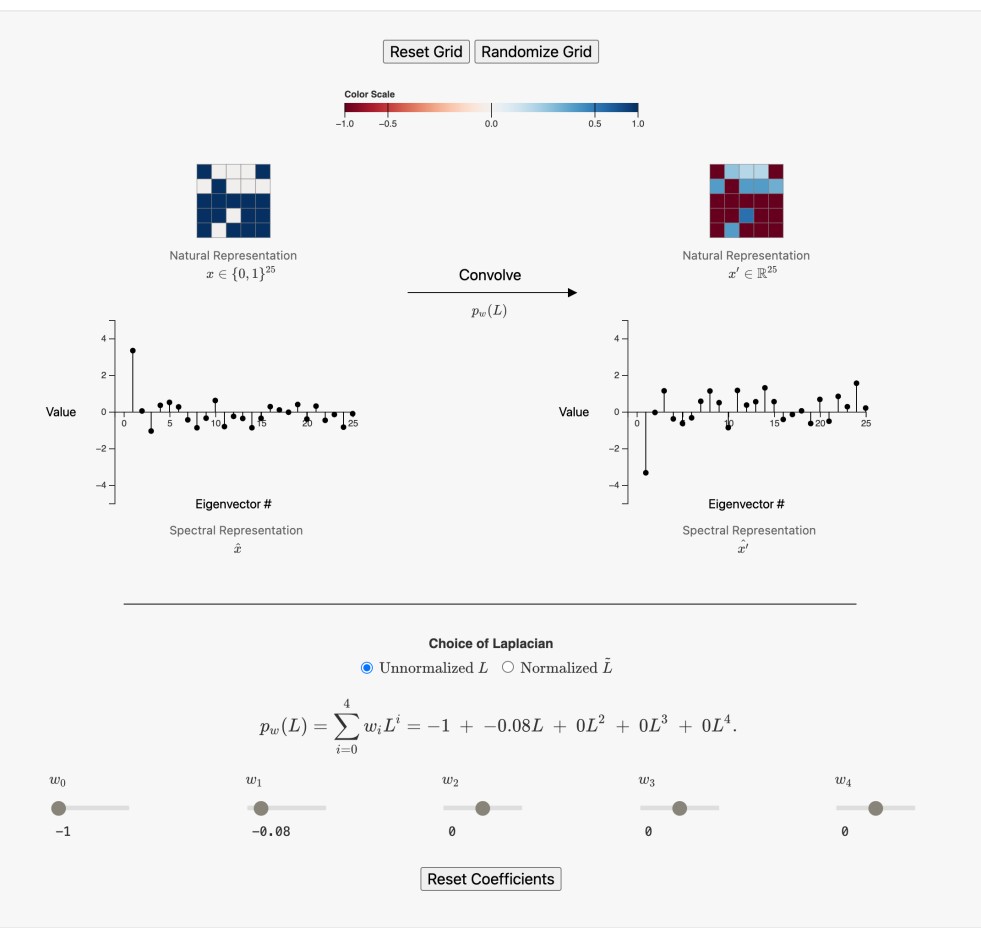

Figure 3: Readers can change the sliders representing polynomial coefficients, and click on the input grid (top left) to toggle binary node features, to see how the output of the convolution (top right) changes.

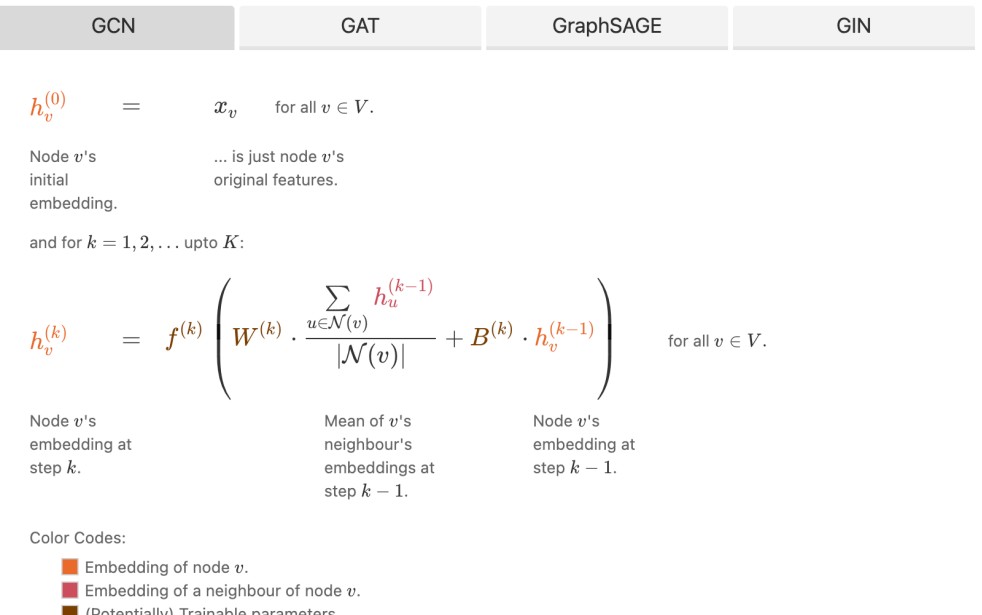

Figure 4: Annotated update equations for different models with a consistent color scheme to identify common characteristics across them.

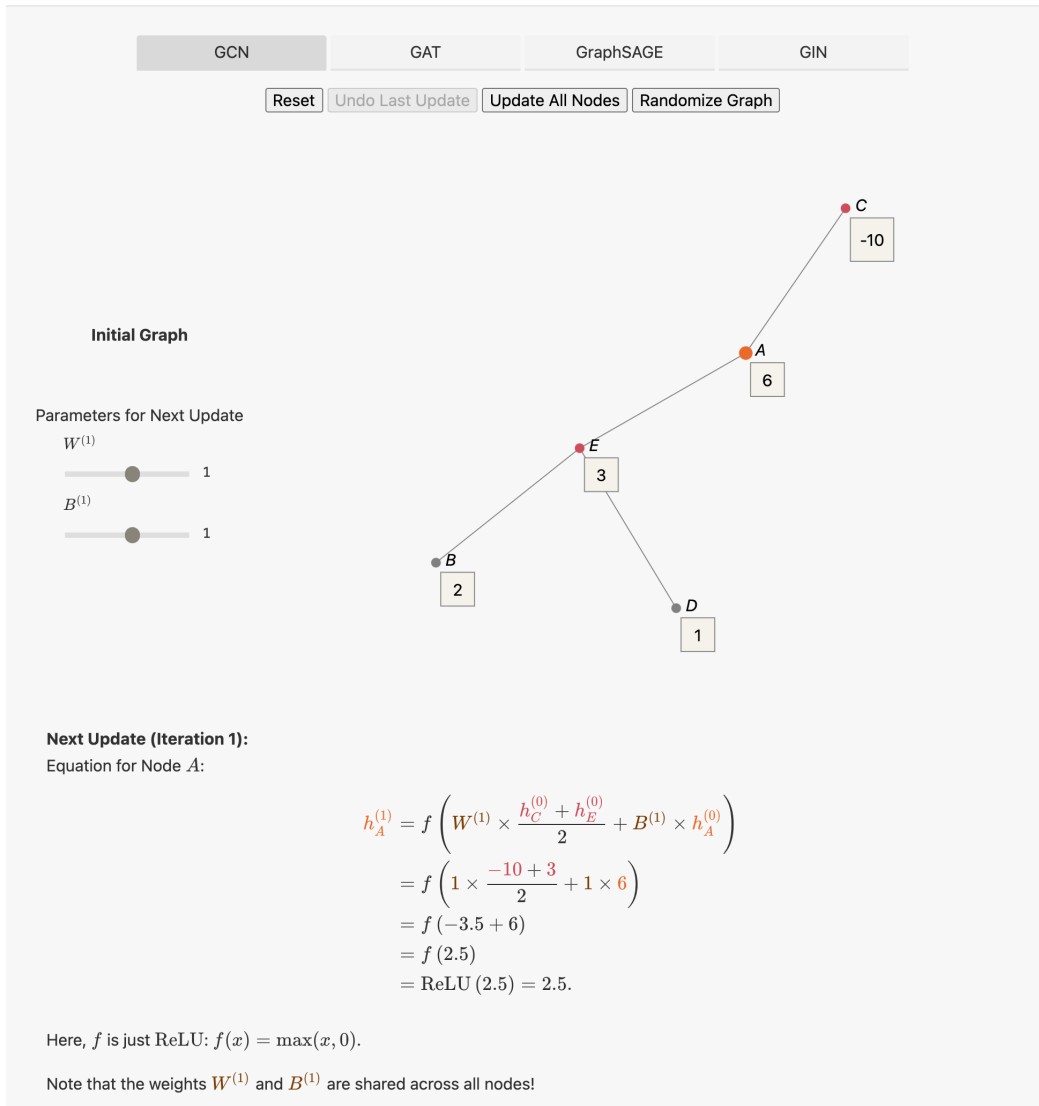

Figure 5: Readers can interact with different GNN models by varying parameters and connectivity of the underlying graph, and seeing how the update equations at each node changes.

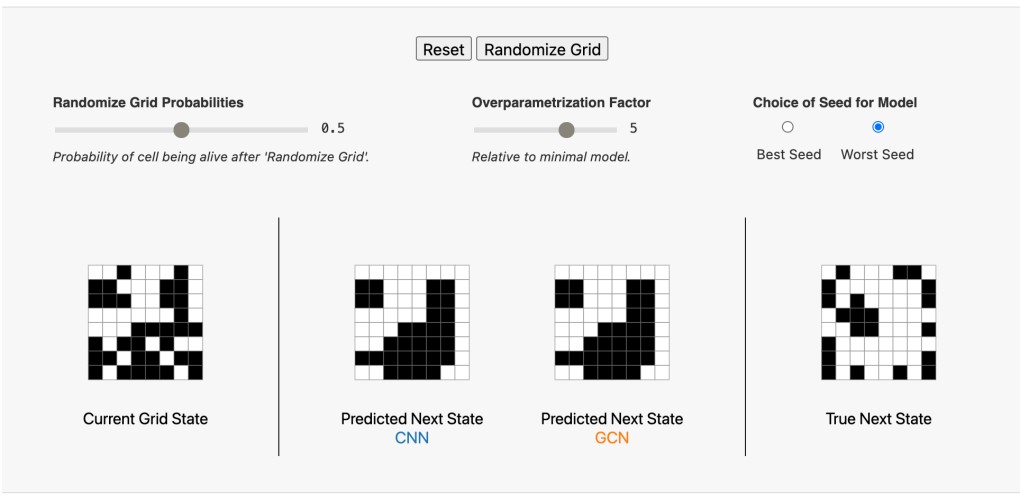

Figure 6: Comparing CNN and GCN models at different levels of overparametrization. Readers can update the input grid (leftmost) by clicking on the individual cells to toggle them, and see how predictions (middle) change.

## 3 CHALLENGES

Here, we talk about the challenges we faced as creators of interactive content:

- **The lack of established high-level libraries to build interactive visualizations.** Authors often need to write their own (often un-reusable) code for generating visualizations as they want to. Here too, we have coded all of our visualizations using the popular low-level JavaScript library d3.js, but we often wished for an interface where we could define high-level objects (such as graphs) and operations on them, instead of creating all of the necessary building blocks from scratch. To help others interested in creating such visualizations, we have created ObservableHQ notebooks, which can be modified and embedded into any webpage, emphasizing the benefits of code reuse.

- **The need for familiarity with web-development technologies.** This article would not be possible without understanding how content is rendered and displayed in the user's browser. With interactive elements, authors need to ensure that user input can be captured without being frustrating to use. It is generally required for modern webpages to be responsive in nature: they must be appropriately viewable at all screen sizes and resolutions. Webpages need to be designed to work just as well across different browsers. Figuring out how to meet all of these constraints can be challenging to authors who have never worked with these tools before.

- **The process of designing visualizations themselves.** As authors, we often deliberated on how much control to provide the readers with for our interactive tools. Adding too many inputs can confuse the reader, especially given the somewhat limited input options (sliders, buttons, checkboxes) that modern browsers provide. Adding too few reduces the flexibility of the tool, and can leave the reader feeling excessively restricted. Performing case-studies to identify the right 'amount' of interactivity can be helpful for more expansive tools, if time-consuming.

- **The process of converting models to run in the browser.** Generally, models are defined and trained using high-level frameworks in Python, which cannot be immediately deployed to the browser in JavaScript. TensorFlow provides a convertor wizard, which eases the process, but authors still need to write code to allow user input, process these inputs and return outputs from the model - all of which are additional non-trivial steps that authors need to spend effort on. These steps would look very different if, for example, PyTorch was used instead of TensorFlow, indicating the need for a framework-agnostic model conversion and execution library.

Dealing with these major challenges is necessary before interactive articles can truly become widespread. We believe that solutions that help authors (such as better input components) are likely to benefit readers in their learning experience, as well.

## 4 ACCESSIBILITY

To evaluate the accessibility of our exhibit, we have used the Web Content Accessibility Guidelines (WCAG) 2.1, which have three levels of accessibility (A, AA and AAA). While our exhibit has been completely validated for W3C compliance[5], there are some areas that still need improving to meet our target of the AA level:

### CRITERIA 2.1.1: KEYBOARD

Certain visual components require clicks via a mouse. We are working on enabling keyboard access for these components. Currently, we have provided 'Randomize Inputs' in these components which allow for the browser to provide these inputs on behalf of the reader.

---

[5]We used https://validator.w3.org/nu/ for checking W3C compliance.

CRITERIA 2.4.7: FOCUS VISIBLE

Certain sliders don't get highlighted when they receive focus. However, each slider is annotated with text which does change when the slider is moved.

## 5   FINAL THOUGHTS

Finally, we analyze the medium for interactive articles on the web, from various perspectives:

- **Presentation:** Interactive articles, if designed correctly, can often convey significantly more information than elaborately designed static content. By allowing readers to interact independently with inputs, and see the outputs for themselves, interactive articles facilitate the understanding of difficult concepts. The helpfulness of interactivity for students in the learning process has already been studied in a formal setting (see Barker (1994) and Laurillard (2013)): our exhibit is a push in the direction to make interactive articles themselves seen as valuable research products.

- **Review cycle**: Interactive articles can be reviewed and edited similar to 'ordinary' research articles, as best shown by Distill, an online journal that publishes many interactive articles, which uses GitHub for both version control and conversations between reviewers and authors. If designed correctly, interactive articles should not be more difficult to use for reviewers not familiar with the new medium, but this requires authors to consider their design choices and implementations carefully. While Distill uses a single-blind review system, it is definitely possible to ensure fully anonymous peer reviews, with additional effort on the part of the authors to ensure the additional tools and services they use (such as ObservableHQ here) do not reveal their identity. However, there are often certain unique characteristics of visualizations designed by a specific author (such as the color schemes, the styling of components, the way user interaction is designed), which are not otherwise apparent when dealing with static diagrams. This can potentially create an additional challenge with interactive articles: leakage of author identity, which can break anonymity in the peer review process.

- **Bibliometrics**: Interactive articles that are officially published can be cited just as any other research articles. What about unpublished 'draft' work that is still relevant to the research community but will need time to be vetted by the peer review process? For 'traditional' research articles, this need is fulfilled by online repositories such as arXiv which provides access and cite-ability to unpublished content in the form of static PDFs. A similar service for interactive articles would be very useful for the field as a whole.

- **Interoperability**: Interactive articles can be snapshotted to get a static view of the article, if absolutely needed. Here, we have used ObservableHQ notebooks to embed our interactive content. This gives us the immensely useful ability to embed these visualizations from the article into any other webpage: enhancing their scope beyond the article itself.

- **Durability**: Interactive articles, for now, remain tied to web services. While it is unlikely that these services will vanish, given their increasing adoption across the world, an archival system (such as arXiv above) independent of publishing bodies, can help alleviate concerns of interactive content simply vanishing.

- **Limitations**: Beyond the challenges for authors listed out in the last section, interactive articles, such as ours, unfortunately remain accessible only to those with access to the web. Identifying a mechanism to convert interactive article into static content with equivalent 'explaining power' is an important challenge for the community to deal with. This will allow interactive articles to piggyback on the modes of distribution designed originally for static content. Even augmenting existing systems to support interactive media will enable these articles to reach a much broader audience.

Here, we have taken great advantage of the interactive medium as a pedagogical tool to explain complex GNN concepts. However, interactive articles undeniably require more effort to create, and come with unique challenges of their own. Dealing with these challenges is important for interactive articles to become more widespread and recognized as true research output.

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
