# OpenReview forum: "Interactive Media for Understanding ML Methods: A Case-Study on Graph Neural Networks"
_ICLR.cc/2021/Workshop/Rethinking_ML_Papers/Exhibit_and_Workflow — Rethinking ML Papers - ICLR 2021 workshop Poster_

### Official Review · Reviewer_SMoN · 2021-03-24
**Article promotes interactive media to facilitate explanation of ML ideas**

**Accessibility:**

Score of 5 (Exceptional): Submission identifies and articulates accessibility matters, provides justifications for the proposed paradigm, and declares the limitations.

**Litreview:**

Score of 4 (Strong): The submission directly differentiates itself from previous works and formats.

**Problemstatement:**

Score of 4 (Strong): The submission sets a very strong example of how to address the problem, which should be relevant to the workshop themes.

**Relevance:**

Score of 4 (Strong): The submission directly addresses a theme of the workshop, and does so in a very professional manner.

**Results:**

Score of 4 (Strong): Submission is very well structured and follows all the criteria (i.e. clarity, novelty, interactivity, and coherency). However, practical significance/theoretical implications are not discussed.

**Reviewerconfidence:**

This is the first paper that I have read that explores the question of what is the best way of presenting scientific ideas. Yet, based on my experience in writing and reviewing ML articles in general, I am fairly confident that the ideas presented in the paper could be useful for the community. I would rate my confidence level as 4.

**Reviewtext:**

Using an example of graph neural networks, the article demonstrates how interactive media can make it easier for readers to understand complicated concepts. The reviewer is convinced by the authors' main points about the potential benefits of such presentation of scientific material. Additionally, the authors also point out potential limitations of the approach - thus, nicely rounding off the article and making it an excellent fit for the workshop.

The reviewer encourages the authors to make publicly available, the tools and code used in the making of their distill article, so as to make it easier for others in the community to also make similar contributions.

**Score:**

Strong accept: The reviewer has a strong enthusiasm to apply the proposed framework in their work.

---

> ### Author Response · Authors · 2021-04-16
> **Updates!**
>
> Thank you for the review!
> Actually our code to create the visualizations has already been open-sourced, but they were listed in the Distill exhibit itself. We have added a footnote indicating where to find the interactive JavaScript notebooks as a footnote on Page 2 now, for clarity.

---

### Official Review · Reviewer_7ZRS · 2021-03-31
**Good fit for the workshop**

**Accessibility:**

Score of 5 (Exceptional): Submission identifies and articulates accessibility matters, provides justifications for the proposed paradigm, and declares the limitations.

**Litreview:**

Score of 2 (Needs Improvement): The submission leaves out prominent examples of previous work in the area.

**Problemstatement:**

Score of 4 (Strong): The submission sets a very strong example of how to address the problem, which should be relevant to the workshop themes.

**Relevance:**

Score of 4 (Strong): The submission directly addresses a theme of the workshop, and does so in a very professional manner.

**Results:**

Score of 5 (Exceptional): Submission has an excellent design and all criteria are addressed. Conclusions, practical/theoretical implications are well articulated.

**Reviewerconfidence:**

3

I believe this work is a good fit for this workshop. I have very limited experience with Graph Neural Network and cannot assert how innovative these visualizations may be, but the references suggest this work is an appreciable improvement.

**Reviewtext:**

This work presents an interactive solution to visualize Graph Neural Network.

The problem is well stated and the visualization briefly explained with several examples. Although the paper lacks explanation on the visualizations there is lengthy discussion about their limitations and implications. The distill exhibit is not part of the paper itself but serves a good detailed explanation of the visualizations.

**Score:**

Accept: The reviewer believes the submission provides a novel and reliable scheme to improve science communication but needs improvement.

---

> ### Author Response · Authors · 2021-04-16
> **Updates!**
>
> Thank you for the review!
> We have added a paragraph + references on research for visualizing classical graph algorithms (VisuAlgo) as well as visualizing graphs themselves (graph layout algorithms). We could not find any prior work that visually describes GNNs however.

---

### Meta-Review · Program_Chairs · 2021-03-31

**Recommendation:** Accept
**Confidence:** 4

**Metareview:**

This is a well-thought-out writeup using GNNs as a running example. I felt the exhibit did try to be in line with what the authors are trying to convey.

I appreciate the authors' efforts on trying to address various aspects and implications of the topic in discussion. However, I also felt the lack of literature review as one of the reviewers felt. I would also urge the authors to make the tools public.

The visualizations are very clear and people should be made aware of making these things with ease. I support accepting the paper.

---

### Decision · Program_Chairs · 2021-04-01

Accept (Poster)